# Wind Energy Conversions, Controls, and Applications: A Review for Sustainable Technologies and Directions

**M. A. Hannan** [1,*], **Ali Q. Al-Shetwi** [2], **M. S. Mollik** [3], **Pin Jern Ker** [4], **M. Mannan** [1], **M. Mansor** [1], **Hussein M. K. Al-Masri** [5] and **T. M. Indra Mahlia** [6]

1 Department of Electrical and Electronic Engineering, COE, Universiti Tenaga Nasional, Kajang 43000, Malaysia
2 Electrical Engineering Department, Fahad Bin Sultan University, Tabuk 47721, Saudi Arabia
3 Department of Mechatronics Engineering, International Islamic University Malaysia, Kuala Lumpur 50728, Malaysia
4 Institute of Sustainable Energy, Universiti Tenaga Nasional, Kajang 43000, Malaysia
5 Department of Electrical Power Engineering, Yarmouk University, Irbid 21163, Jordan
6 Centre of Green Technology, Faculty of Engineering and Information Technology, University of Technology Sydney, Ultimo, NSW 2007, Australia
* Correspondence: hannan@uniten.edu.my

**Abstract:** The use of renewable energy techniques is becoming increasingly popular because of rising demand and the threat of negative carbon footprints. Wind power offers a great deal of untapped potential as an alternative source of energy. The rising demand for wind energy typically results in the generation of high-quality output electricity through grid integration. More sophisticated contemporary generators, power converters, energy management, and controllers have been recently developed to integrate wind turbines into the electricity system. However, a comprehensive review of the role of converters in the wind system's power conversion, control, and application toward sustainable development is not thoroughly investigated. Thus, this paper proposes a comprehensive review of the impact of converters on wind energy conversion with its operation, control, and recent challenges. The converters' impact on the integration and control of wind turbines was highlighted. Moreover, the conversion and implementation of the control of the wind energy power system have been analyzed in detail. Also, the recently advanced converters applications for wind energy conversion were presented. Finally, recommendations for future converters use in wind energy conversions were highlighted for efficient, stable, and sustainable wind power. This rigorous study will lead academic researchers and industry partners toward the development of optimal wind power technologies with improved efficiency, operation, and costs.

**Keywords:** wind energy conversion; converter controller; maximum power point tracking; future converter technologies

## 1. Introduction

Energy is a fundamental component of our existence and the foundation of civilization. In most circumstances, the contemporary world's social and economic well-being relies on the availability of renewable and sustainable energy [1]. In contrast, the rapid and out-of-control expansion of human civilization and industry has a significant detrimental impact on both the environment and the world's limited supply of fossil fuels. Sustainable development concepts and criteria must be used in technical processing, products, and activities going forward if we want to stop additional environmental harm and the loss of our natural resources [2]. Thus, more attention is being paid to developing renewable energy sources. In the upcoming years, the world's capacity to produce electricity through the use of solar power, wind power, and other renewable technologies is expected to increase [3].

Due to technical and economic visibility, wind power has emerged as one of the most promising renewable energy sources (RES) in recent years [4]. Wind energy capacity can range from a few kilowatts to many megawatts and can be found in many applications [5,6]. Wind energy can be used in both minor off-grid systems and substantial wind farms connected to the grid. This sort of distributed generation poses issues with the interconnection system due to the absence of active and reactive power control. Consequently, this approach necessitates careful control, modeling, and choosing a suitable wind power system. The widespread use of wind power has been directly tied to the development of the technology for wind turbines and control technologies over the previous two decades [7]. In this regard, about 102 GW of wind power capacity was added in 2021. Annual increases raised the total capacity by 13.5%, reaching more than 845 GW. In terms of market share, China dominated the market, followed by the USA, UK, Brazil, and Vietnam [8]. Figure 1 indicates growth in installed global wind capacity to approximately 845 by the end of 2021 compared to 651 GW in 2019 [3]. It is expected that installed wind capacity will increase dramatically over the next few years due to the continuing desire for alternative energy sources.

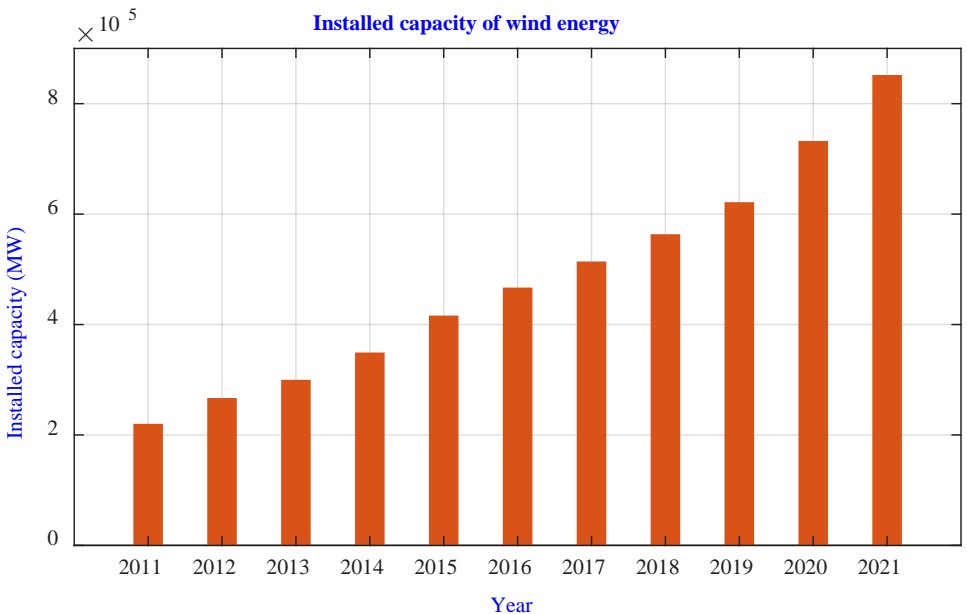

**Figure 1.** Installed capacity of wind power during recent years.

The primary goals of the control strategies for better wind energy operation are to reduce dynamic and static mechanical loads, offer stability for grid integration, maximize power generation, and ensure a reliable grid [9]. In this regard, power electronics (PE) play a significant role in wind systems' efficient control and optimal operation [10]. The converter technology used in wind power applications has changed significantly over the past several years due to wind turbine systems' (WTS) rapidly increasing capacity and increasingly important effects on the electrical grid [11]. Converters continuously develop, resulting in notable performance enhancements for wind turbines that not only lower mechanical stress and boost energy output but also allow the entire wind turbine (WT) to function as a fully controllable power source, significantly improving the integration of wind energy into the power grid [12]. One of the most prevalent wind turbine types is the doubly fed induction generator (DFIG). This type of wind turbine comprises a slip-ring induction generator, a partial-scale electronic power converter, and a common capacitor in the DC link [13]. In such a situation, dual-feed asynchronous generators are receiving more and more attention in the wind power conversion system. Because with commercial two-way pulse width modulation (PWM) inverters, it is possible to regulate the rotor current via field orientation control, resulting in a decoupled control of stator-side reactive

and active power, resulting in a tiny portion of the overall system's power being processed by the power converter.

Despite the incredible expansion of wind energy, researchers face numerous hurdles, including the unpredictability of the wind's nature, grid connection, and the positioning of wind turbines [14]. To integrate wind turbines into the electricity system, more sophisticated contemporary generators, power conversion devices, energy management, and controllers must be advanced [12]. According to the recent available literature, it is observed that many papers have reviewed the wind energy system development from different aspects such as the wind turbine reliability [15,16], operations and maintenance [17], reliability and cost of energy [18], design [19], control [20,21], damage detection techniques [22], noise [23], and maximum power point (MPPT) control methods [24–26]. However, a comprehensive review of the PE role in the wind system's power conversion, control, and application towards sustainable development has not been thoroughly investigated. Thus, this paper proposes a widespread review of the impact of converters on wind energy conversion with its operation, control, and related issues. This article's objectives and contributions can be summed up as follows:

- Investigates the prospects and recent advances of converters' contribution towards efficient wind energy conversions.
- Comprehensive analysis of the trends and diversity of converters in wind power: operations, topologies, applications, challenges, and future prospects.
- A comprehensive discussion outlines wind energy advancement in terms of the control system, main features, and related applications.
- Finally, based on the review, recommendations for future improvement in the performance of wind energy-based converters are highlighted for a sustainable future for the wind energy system.

## 2. Reviewing Process

After a rigorous study of various articles, a review methodological framework has been developed to comprehensively review the converter contributions towards modern wind turbine controllers and their integrations. The challenges and factors that substantially affect the performance of wind energy-based converters are also identified. In this stage, three screening and assessment phases were employed to select a suitable number of works of literature. Subsequently, 533 papers were determined following the preliminary screening. The article selection through the second screening phase was performed using the essential keywords, including wind energy conversion system, wind power, pitch angle controller, and future converters. A total of 287 papers were found following the second screening, in which the paper title, abstract, subjects, and contributions were evaluated to explore the relevant articles for this stage. The article's final selection is carried out through the citations, impact factor, and review process. In sum, the review, analysis, and critical discussion relating to converters' contribution towards wind energy conversions, controls, and applications, along with issues and challenges, are conducted using the final filtered 170 articles. The methodological framework is shown in Figure 2.

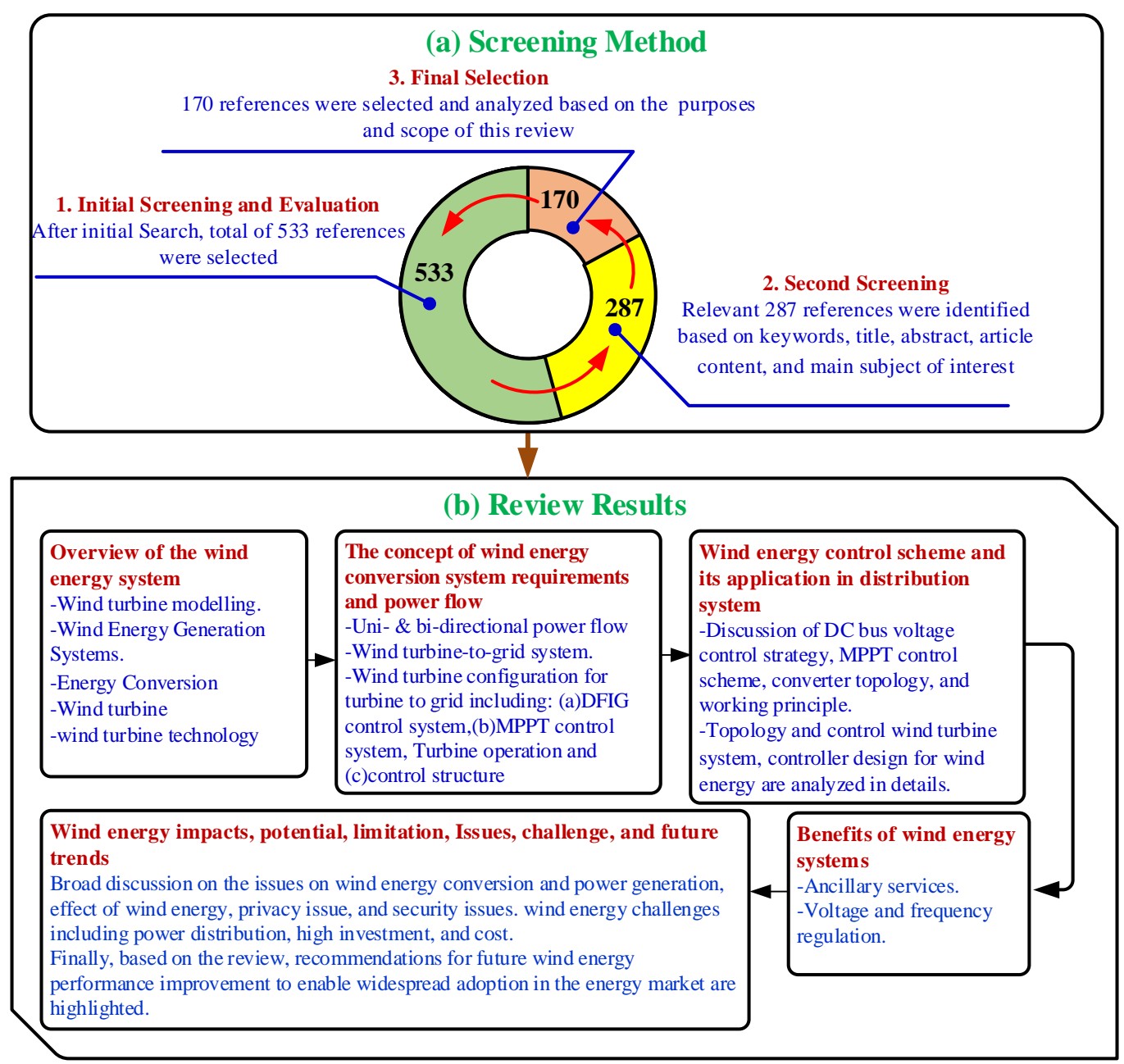

**Figure 2.** The article selection procedures.

### 3. Wind Energy Conversion System

The wind energy conversion system (WECS) contains wind turbines and converter converters. Using wind turbines to extract the wind's mechanical energy, the generators convert it into electrical energy, and the converter system is in charge of transferring the generated energy to the power network or a battery bank. When converting wind energy to electricity at a variable speed, the most commonly utilized generators are synchronous and doubly-fed induction generators (DFIG) [27]. When using induction generators, the rotor and stator are both linked directly to the network, but the electronic converter acts as a mediator between the two. Since the rotor circuit may independently change the amplitude and frequency of the produced voltage, the DFIG has long been the preferred choice for large, variable-speed WECS that are connected directly to the electrical network, as illustrated in Figure 3a [27–29].

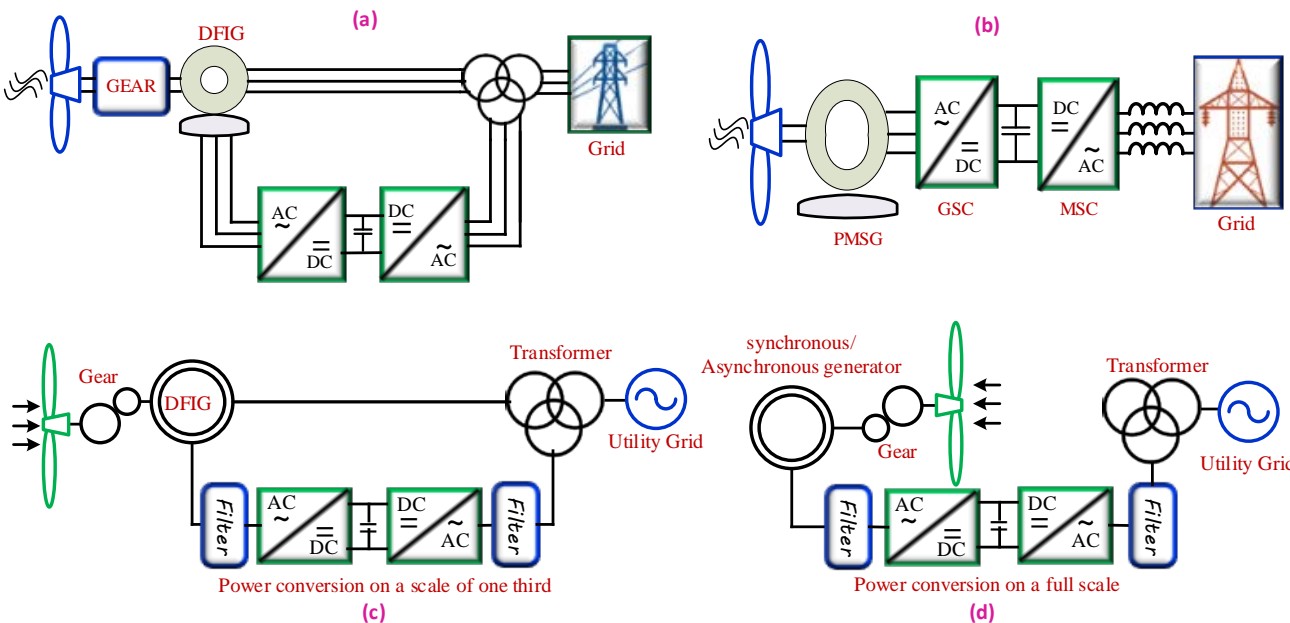

**Figure 3.** Partially rated converters conversion system and full-scale power converter of variable-speed: (**a**) partially rated converters conversion system; (**b**) converters conversion system on a full-scale more significant level; (**c**) a wind turbine with a variable speed, DFIG and partial-scale power converter; and (**d**) full-scale variable-speed WT power converter.

Generators that generate synchronous power are disconnected from the power distribution grid [28]. As a result, they are included in systems that utilize full-scale converters. Due to the lack of a gearbox, the multipole permanent magnet synchronous generator (PMSG) has been the preferred alternative in recent years, reducing WECS losses, reducing maintenance requests, and enhancing system efficiency and dependability [30]. Figure 3b illustrates its configuration [27]. Dc-links frequently employ a chopper circuit to disperse power in the event of grid disruptions [31]. When compared to DFIG, PMSG offers the following advantages: no external stimulating current is required; high reliability; lightweight; low maintenance; small size; and high efficiency [29]. The fundamental elements of a WTS's control system, including the turbine, generator, filter, and converter, as displayed in Figure 3a–d [32]. It is possible for the designed wind turbine to either be of the kind seen in Figure 3c or of the kind depicted in Figure 3d [32,33].

Previously, the usage of permanent magnet synchronous generators was recognized primarily in tiny WT but not in large-scale power generation due to the huge and heavy permanent magnets required [34]. The usage of wind turbines based on permanent magnet generators (PMG) is quickly increasing because of the advancement of semiconductor switches and the improvement in efficiency and reliability; also, the innovation of materials utilized in the rotor of the generator has permitted the use of PMG at high power [35]. For example, a permanent magnet-based-high-temperature superconductor is used in the rotor to achieve higher magnetic density: a 15-mm thick segment of permanent magnets can generate the same magnetic field as a 100–150-mm section of copper windings. Furthermore, the setting up of a gearbox, which is mandatory for large and medium WT, can be avoided using direct drive variable speed. Due to its simplicity, the direct drive wind turbine with PMG is now employed in the wind power system as the most promising one [35,36].

### 3.1. Wind Turbine Operation

The WECS uses improved control algorithms due to the rapid advancement of industry expertise [37]. In a WECS, the wind's kinetic energy is transformed to mechanical one using the WT, which is subsequently transformed into electrical energy. Because wind power is not ready to be integrated into the grid, several converter topologies have been developed

to properly govern the grid-side converter (GSC) and the machine-side converter [38]. As a response, the fundamental control of WECS is used to serve the electricity network's needs at different wind speeds, as presented in Figure 4a. The unpredictable nature of the speed of the wind and the variability of the climate highly influences wind energy's dependability. Because of this, it is essential to understand the nature of wind and identify its operating areas to effectively integrate the WTs into the utility grid according to the speed measured [39,40]. Thus, for a particular range of the speed of wind constrained by cut-out ($V_{\text{cut-in}}$) and cut-in ($V_{\text{cut-out}}$) speeds, WT can be used to harvest the accessible wind power, as explained in Figure 4a,b [39,40]. The typical variable and fixed wind speed power curves are shown in Figure 4b.

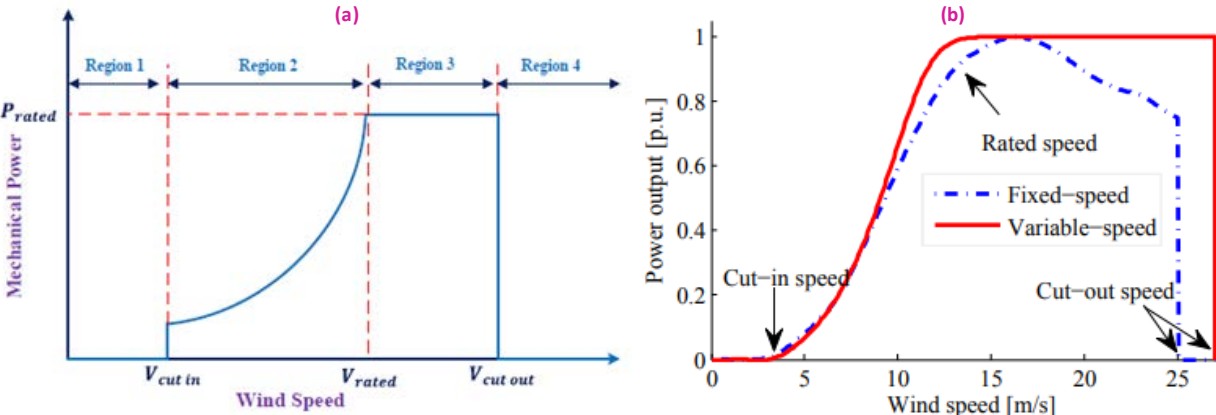

**Figure 4.** (**a**) WT operating regions. Reprinted with permission from Ref. [41], (**b**) typical variable and fixed wind speed power curves [42].

Accordingly, the following is a breakdown of WT's operational areas:

- Regions 1 and 4: As a result of safety concerns, it is necessary to stop and disconnect the WT from the electrical power network.
- Region 2: Wind power is harvested to its fullest potential in the central region using MPPT systems.
- Region 3: WT blades are protected from mechanical stress at high wind speed by limiting the generated power to its rated output through pitch angle control.

The WTs can function as a fixed-speed WT (FSWT) or a variable-speed WT (VSWT) [42]. As a result of their limited speed range and high mechanical stress, FSWTs face a number of major issues. On the other hand, The VSWTs, are used to address the limitations outlined above [43]. The VSWT can run to capture the most electricity at every wind speed, decreasing mechanical stress on the WT and minimizing power variations, which reduces mechanical stress on the WT [44,45]. Rotor speed can also be continuously varied to keep a constant ratio of rotor speed to the speed of wind in response to instantaneous variations in wind speed. In the absence of a consistent ratio, wind power extraction will be reduced to a minimum [39,40,46].

### 3.2. Wind Turbine Configurations

In a WT, the kinetic or mechanical energy of the wind is converted into electricity. When connecting WTs to electrical grids, three primary configurations may be utilized for this purpose. An induction machine is a typical cage-rotor induction generator that is promptly linked to the utility grid without needing a power exchanger. Reactive power is necessary for the induction machine to work. Both the utility power system and machine-terminal capacitors can be used for this purpose. These devices are incapable of delivering any reactive power [47]. DFIGs are used in the second kind of design, which involves the usage of a wound-rotor. Slip rings are used to capture electricity from a rotating rotor at a slip frequency. This ac power must flow via a converter-based rectifier and inverter

system to be converted to a voltage and frequency compatible with the electric power system. Because of this configuration, the size winding of the stator generator can be reduced by 25–30%, with the converters compensating for the power discrepancy between the generator and rotor power. On the other hand, the expense of converters increases the overall cost of such a system [19]. Variable voltage and variable frequency outputs are produced using a permanent or conventional magnet synchronous generator in the third kind of WT architecture. Therefore, an inverter and rectifier based on power electronics are required to convert the WT's total rated output power to a level compatible with the utility power grid [48]. The two more contemporary designs (both of which feature converters) enable the wind turbine to function in a variable speed mode, which can enhance the total amount of wind power collected by the turbine [19,47].

There are several ways to convert wind power into electricity, but the induction generator is the most popular choice. Wind power generation using a squirrel cage induction generator (SCIG)- is one of three primary wind farm (WF) designs now in use [49]; a wind energy system using both a DFIG and a directly driven synchronous generator (DDSG) is explained in Figure 5 [50]. Figure 5a shows the WFs with SCIG, which is the most cost-effective approach because it is linked directly to the electrical network. It is common for a capacitor bank to be put at one of the wind turbine's terminals to recompense the local reactive power created by the wind turbine, which fluctuates depending on how much power is generated [50]. Back-to-back converters that only have a portion of their power supply are used to separate the frequencies of the mechanical and electrical rotors, as seen in Figure 5b. Last but not least, the WF with DDSG is shown in Figure 5c, where full power back-to-back (b2b) converters are utilized to disconnect the generator from the utility grid completely. In new WFs, it is becoming increasingly popular to have DFIG or DDSG linked to the grid through a b2b converter. However, half or more of the currently deployed WFs continue to use the SCIG architecture [50,51].

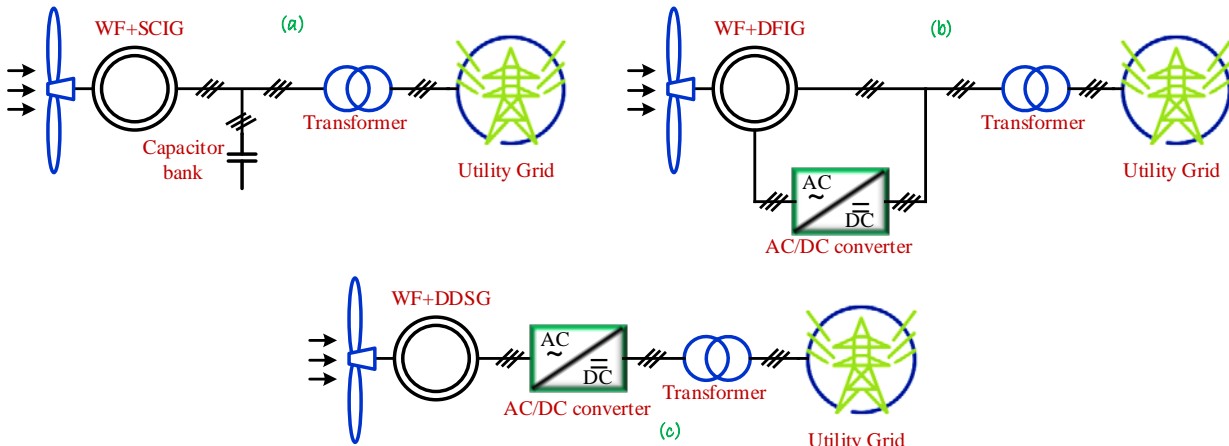

**Figure 5.** WFs equipped with an induction generator and a standard transformer for interfacing (**a**) SCIG. (**b**) DFIG. (**c**) DDSG.

There have been many advancements in wind turbine technology since the early 1980s, but the basic concept has remained the same. To control the linked shaft speed, a horizontal rotor with three pitchable blades is used in the modern horizontal-axis wind turbine (HAWT) to harvest wind energy. This type of rotor has three blades and is widely used. It usually has a front bearing that is independent and is connected to a gearbox at a low speed, making it appropriate for the most common types of four- and two-pole generators [52]. On the vertical-axis wind turbine (VAWT), the shaft's axis is vertical and perpendicular to the ground, allowing it to rotate freely. When the wind is blowing, VAWTs always face the wind. In general, the modern wind industry faces a major problem in designing the most efficient wind turbines to harness wind energy and produce electricity [53]. It has taken the wind turbine industry 30 years to go from an idealistic fringe activity to a major player in

the power production industry since the turbines have grown in size by a factor of 100 and energy costs have decreased by more than 5. Simultaneously, the engineering foundation and computational tools have evolved to accommodate the machine's scale and volume. This has been a great wind turbine narrative up to this point; nevertheless, there are still many technological obstacles to overcome and many more spectacular feats to come [54].

## 4. Wind Energy Converters, Their Issues, and Integration

As a means of controlling and decoupling the wind turbine generator from the electrical grid and improving dynamic and steady-state performance, converters are central to several potential solutions of a technical nature for the electrical systems of wind turbines [55]. The most significant converter applications in WTs are the focus of this section.

### 4.1. Soft-Starter for Fixed-Speed Wind Turbines

Directly linking a WT to the electrical grid, often known as the "Danish concept" [56,57] was a common practice in early WT systems. The SCIG is linked to the grid using a transformer, and its speed of operation is very close to being constant. There are various aerodynamic approaches to reducing the engine's power output, such as stall control, pitch control, and active stall control. Figure 6a represents the fundamental combinations that make up the fixed-speed ideas with soft starter [57,58]. Induction generator-powered WTs have the advantage of being simple and affordable to construct without requiring a synchronization mechanism. These solutions are appealing owing to their low cost and dependability. However, it has some drawbacks, such as (*a*) constant speed is required for the wind turbine to function; (*b*) a strong power grid is needed to maintain a steady functioning; and (*c*) there may be a need for an additional, more expensive mechanical structure to handle the additional mechanical stress as a consequence of wind gusts on the drive train [57].

A direct connection between an induction generator and a wind turbine's induction generator creates transients with high inrush currents that disrupt the grid and cause elevated torque spikes in the wind turbine's drive train. Such a transient disrupts the system and restricts the total of WTs that may be installed. Soft-starting thyristors are frequently utilized to minimize the high starting currents of induction generators [59]. Typically, based on the thyristor's technology, the soft starter or current limiter restricts the inrush current's value to less than twice the generator's rated current. The soft-starter can only handle a certain amount of heat, and when the connection to the grid is finished, a contractor that carries the full-load current shorts it out [60]. However, without a soft–starter, the current will stay high at its peak value (1 pu), as seen in Figure 6b [61]. In sum, with the help of a soft starter, peak currents are effectively dampened, reducing the strain on the gearbox. This decreases the impact on the grid and the accompanying costs [59,60].

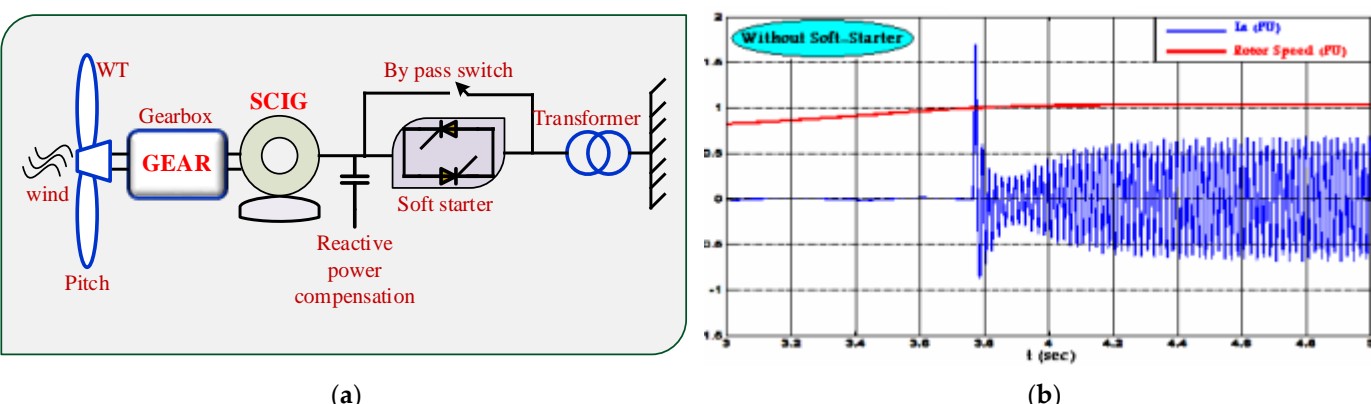

(**a**)                                                         (**b**)

**Figure 6.** (**a**) Fixed-speed WT built on a cage-induction generator with a PE soft-starter, (**b**) the behavior of the current (*Ia*) without a soft starter [61].

### 4.2. Variable-Speed Wind Turbine Control

Many benefits may be gained from operating a WT system at variable speeds. Wind turbines, for example, can rise or reduce their speed depending on the torque and wind speed. Consequently, the gearbox, tower, and other drive-train apparatuses will last longer [62]. In addition, variable-speed devices can enhance energy output and minimize the variability of the power pumped into the power network. In variable-speed mechanisms, a converter scheme is used to link the generator to the power network [63]. For induction and synchronous generators with free rotor windings, a full-rated converter system must be installed between the grid and the generator stator, and the system must be fed with the entire power output [64]. The grid is directly connected to the stator of an induction generator with rotor windings, and a rotor that is either connected to a resistor with a converter controlling it or that is linked to the power system using a power electronics converter and slip rings [65]. The grid-connected variable speed wind turbine through an AC-DC-AC converter can guarantee many advantages [33].

### 4.3. DFIG Wind Turbine Converter Control

Both quick and slow control dynamics are involved in the process of regulating a wind turbine [66]. However, the main components of a WTS's control system contain the WT, generator, converter, and filter [32]. Commonly, the power entering and leaving the generation system must be controlled with care. Mechanical components must be used to regulate the turbines' output power (e.g., blade pitch, yaw system, etc.). In the interim, the entire control system must adhere to the energy generation directives issued by the transmission and distribution system operators. Consideration can be given to more sophisticated wind turbine control mechanisms, such as power generation optimization, grid troubleshooting, supporting the grid in both abnormal and normal modes, etc. [31]. The generator's current should normally be controlled by managing the converter on the generator side, allowing the turbine speed to be altered to raise power generation according to vacant wind power [34]. The coordinated control of numerous WT subsystems, including the grid converters, generator, crowbar/brake chopper, and tilt angle controller, is required to operate under grid failure scenarios [67]. Finally, the wind power converter performs fundamental controls such as DC bus stability, current regulation, and grid synchronization as rapidly as possible. PI and PR controllers are the most popular controllers utilized in this context [35]. Figure 7a depicts an example of DFIG-based WTS administration during the abnormal or faults modes in the grid. During any type of fault, the wind turbine should have a protection scheme (known as fault-ride-through) until the fault is cleared based on the scheme shown in Figure 7a.

Recently, it is required from wind energy systems to act as traditional generators when any faults happen in the system, such as double-line-to-ground (2LG) faults. The 2LG is typically a short circuit between two phases (i.e., phases A and B) to a common point, with a fault resistance from the common point to the ground. Figure 7a depicts an example of DFIG-based WTS administration during the abnormal or faults modes in the grid. During any fault, the wind turbine should have a protection scheme (known as fault-ride-through) until the fault is cleared based on the scheme. For this purpose, the authors in [68], proposed the AC-DC unified power quality conditioner to confine the power oscillations and protect the devices from high currents during the fault using the chopper and crowbar circuits. Figure 7b clearly demonstrates that under 100% 2LG fault, the maximum output power (which is 2 pu during normal operation) can be reduced from 3.93 pu to 2.25 pu. In addition, the protection scheme using a chopper and bypass crowbar can limit the high value of current and DC bus voltage during the fault within 2.0 pu and 1.1 pu, respectively, as illustrated in Figure 7b [68].

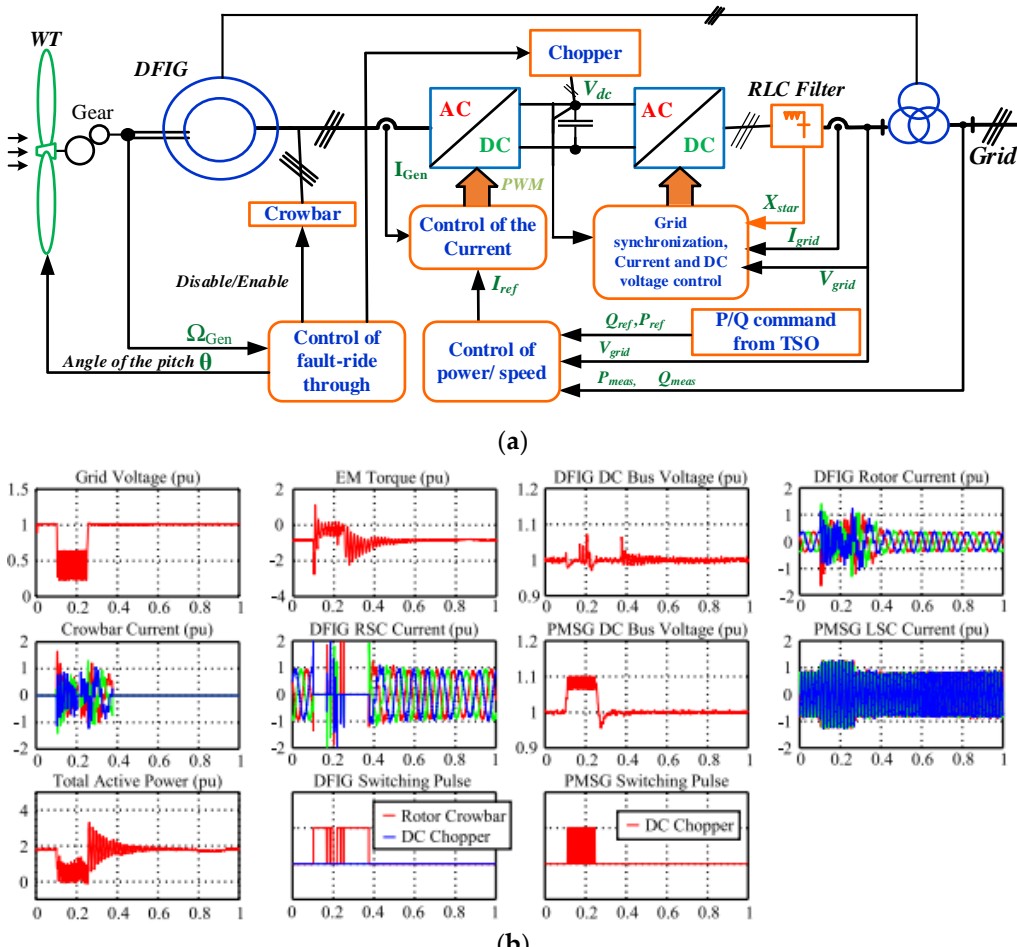

**Figure 7.** (**a**) Control of a wind turbine with DFIG during grid disturbances, (**b**) unified power quality conditioner-based fault-ride-through and protection scheme using a chopper and bypass crowbar during 2LG fault. Reprinted with permission from Ref. [68].

The rotating speed of a wind turbine normally changes in relation to the wind speed and maintains a constant angle of inclination below its increased power output. The rotating speed will be regulated at the maximum permissible slip during very low wind speeds to prevent overvoltage at the generator output. Limiting the turbine output power when it reaches its maximum rated output is done with the help of the tilt controller. The total amount of electrical power the WTS generates is regulated by a converter located on the rotor side of the DFIG [69]. The strategy for controlling the mains-side drive is to maintain the connection's DC voltage at a fixed level at all times [13]. There is a tendency to utilize a crowbar-connected to the DFIG rotor to enhance control efficiency in case of network disturbances, as shown in Figure 7a [70].

DC connection allows some decoupling between the turbine and grid, which is a benefit of this technology. The DC-link will also permit the connection of wind turbines to energy storage devices that can more effectively regulate the flow of active electricity into the grid. This function will significantly enhance wind turbines' capacity to assist the grid [71]. The converter on the generator side controls the active power that is generated by the WTS, while the converter on the grid side controls the reactive power [72]. There should be no doubt that a DC breaker can be used in the case of a mains failure to protect the DC connection from overvoltage if more turbine power has to be dissipated in the form of a quick voltage drop [69,72]. The architecture of the DFIG system is somewhat complicated. A drive train that contains low- and high-speed shaft connects the DFIG to the wind turbine. The windings of the step-up transformer are directly linked to the

windings of the stator of the DFIG's stator. A b2b converters are utilized to establish a connection between the grid and the DFIG rotor windings so that the speed and frequency of the windings can be controlled [73]. The architecture of wind energy conversion using the DFIG technology is shown in Figure 8a.

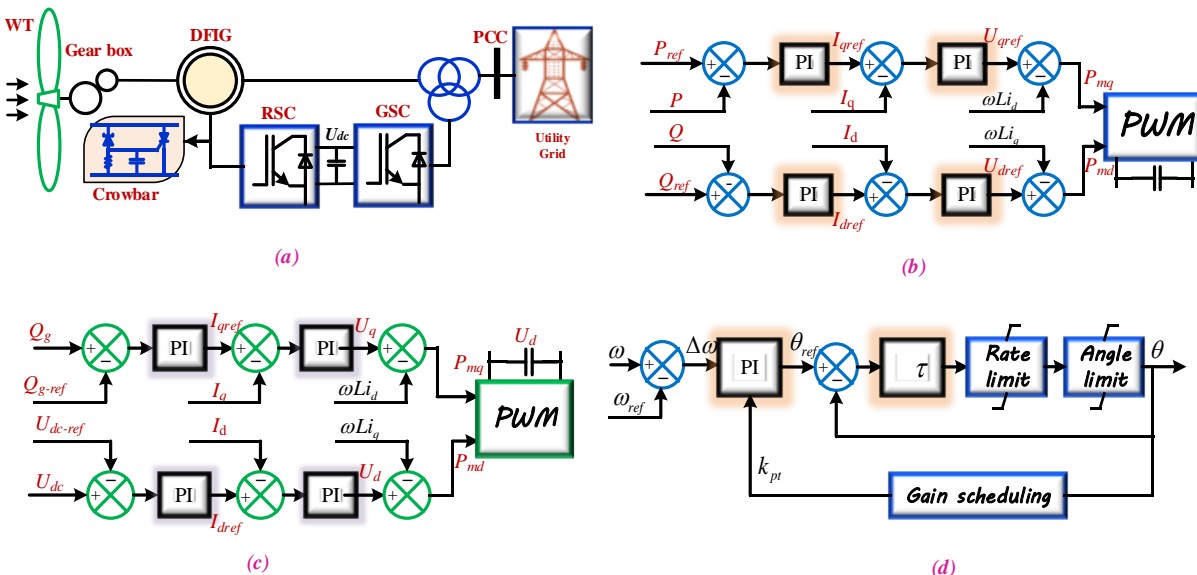

**Figure 8.** (**a**) Architecture of wind energy conversion using the DFIG technology [73]; (**b**) schematic representation of the RSC control block [73]; (**c**) Schematic representation of the GSC control block [73]; and (**d**) pitch angle control.

The b2b pulse width modulation (PWM) converter comprises two pulse-width modulated voltage source converters, the GSC and the RSC. These converters are coupled together via a DC bus. The crowbar protects DFIG and rotor side control (RSC) from harmful inputs such as excessive current and voltage [74]. If the current in the rotor is more than the maximum that is permitted, the DFIG control system may be disassembled into its two essential components, which are the control of the GSC and the RSC. The control of reactive and active power can be kept completely distinct in DFIG wind power systems thanks to vector control [75]. The RSC controller controls the stator's reactive and generated reactive power. A representation of the RSC's vector control system can be seen in Figure 8b [74,75].

A double-loop control approach comprising an outer loop (power) and an inner loop (current) has been implemented. The power control loops generate all the reference values for the present control loop. In this case, specialized MPPT management yields the best active power reference $P_{ref}$ [76]. While contributing to an inefficient electrical system, it is possible to chage the values of the $Q_{ref}$ to be greater than zero so that the RSC can give reactive power in order to keep the voltage stable. The $i_{qref}$ and $i_{dref}$ can be applied in order to acquire $u_{qref}$ and $u_{dref}$, respectively. The control design for the GSC is seen in Figure 8c [77].

GSC's primary function is to control the voltage across the dc connection. It is possible to individually manage active power (DC voltage) and reactive power using the GSC control system, which does so by manipulating the q- and d-axis currents. Reactive reference $Q_{ref}$ is often zeroed out to reduce DFIG's current draw and corresponding losses [78]. In addition, the GSC control system may be configured to respond quickly to the grid's reactive power needs for voltage support. GSC can deliver reactive power even if RSC cannot do so due to the severity of the malfunction [75]. As shown in Figure 8d, the pitch angle control is applied so that the rotor speed does not exceed its rated value. This keeps the rotor from becoming unstable. When the wind speed is higher than its rated value, a PI controller is used to implement the pitch controller in the system. An error signal is produced by

comparing the actual rotor speed with the reference signal when the speed of wind is lower than the rated speed ($\beta=0^\circ$). In contrast, a reference signal is generated when the maximum speed is lower than the wind speed. If the rotor is going too fast, the pitch angle will be increased to keep it under control.

Voltage control, power flow management, damping power oscillations, transient stability, etc., are some of the key operational issues for the modern wind power system [79]. Consequently, reactive power compensators such as static synchronous compensators (STATCOM) are an effective technique for managing the voltage at the connection point [13,79]. Although this is a cost-effective approach, the fact that these WFs require large power transformers for both the STATCOM generators and the WT generators is a drawback of using these WFs [80]. On the other hand, no researchers have looked at the use of solid state transformer (SST) with full use of all functionalities, despite the fact that it is viewed as an innovative way that integrates active power transmission, reactive power correction, and voltage conversion [81]. Consequently, the primary contribution of this paper is the proposal of a novel family of WF designs with an SST interface, which effectively replaces the usual transformer and reactive power compensator.

### 4.4. Sliding Mode Controllers

In general, sliding mode controller (SMC) design strategies can manage nonlinear systems and provide intelligent, resilient responses to unstable systems [82]. In [83], the authors of this study proposed an approach for the design of SMC for use in WT systems that uses a dual-output asynchronous generator connected directly to the grid. An easy-to-use SMC was designed by H. De Battista and his colleagues [84] by interpolating the rotor and stator torque with a simple static converter. In this way, system damping can compensate for the significant loss of generator power and torque variation. The technology is completely resistant to generator clipping faults and AC line voltage disruptions. Furthermore, Baloch MH et al. [85], introduced an ordering alternative switching meaning that based on job stipulations, the firing angle and generator torque characteristics may be adjusted. The simulation results show that power grid disruptions and uncertainty in electrical restrictions are completely resilient.

Menezes et al. [21] established two essential algorithms for miniature wind energy conversion devices to overcome the abovementioned challenges and prevent chattering. The authors improved both integrated variable structure controllers (IVSC) and variable configuration. When the generator speed of the WT was at its highest, the variable structure integral (VSCI) operated well. Although it is worth noting that VSCI performs somewhat better than the other methods since control functions in IVSC techniques need an estimate of the sign function. As a result, an accurate assessment of wind energy transformation systems is required for such a control approach. Sierra et al. [86], studied the use of an SMC for the model of a changeable structure system. Based on a differential geometric approach to non-linear systems, this control strategy excels in dealing with uncertainty and noise. The SMC described by the author is designed to reduce chattering, be simple, resilient, limit disturbances, and achieve mode control. Referring to [87], variable structure control and Lyapunov methods are used in a brushless wind power conversion system with a dual-fed jet machine and a multiple-input-multiple-output control system. The capacity to reduce the amount of chatter and the number of interruptions that occur can thus be attained in this manner. Zuo et al. [88] developed a doubly-fed wind turbine system using a directly controlled matrix converter and sliding mode control techniques instead of employing static converters. It is important to mention that the SMC is developed and adapted using different methods such as fractional-order SMC, high-order SMC, adaptive SMC, fuzzy SMC, and neural network SMC. The distribution of the total number of literature focused on SMC-based control is presented in Figure 9a [89]. Figure 9b,c shows the conventional SMC [90] and artificial neural network (ANN)-SMC [91], respectively.

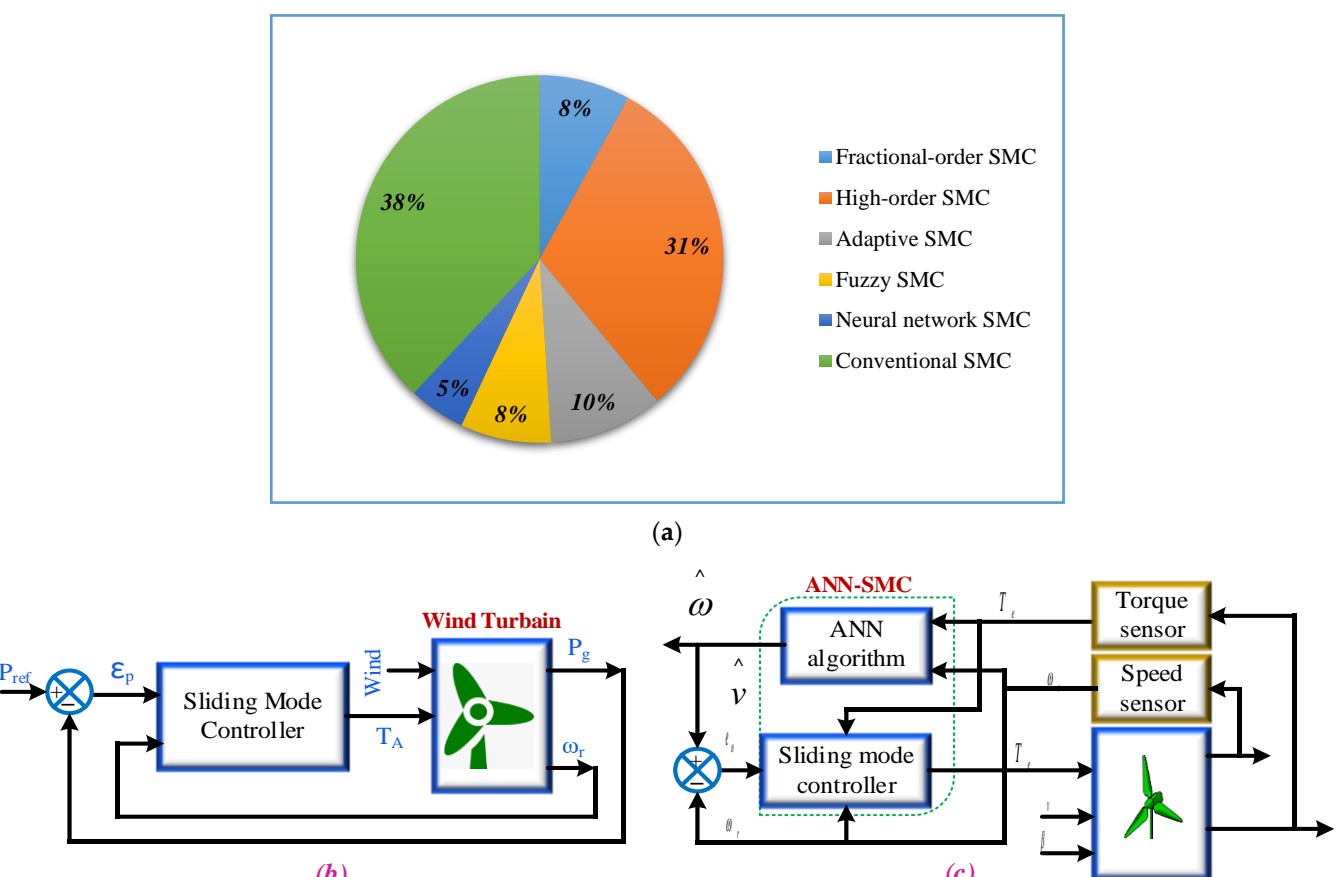

**Figure 9.** (**a**) The distribution of research studies focused on sliding mode controller (SMC)-based control; (**b**) conventional SMC; and (**c**) artificial neural network- sliding mode controller (ANN-SMC).

## *4.5. Predictive Controllers*

Nonlinear control problems with restrictions can be handled quite effectively using the predictive control approach. R. Ruiz-Cruz et al. [92] employed a double-fed induction machine, and anticipatory control approaches were created for a power grid that was coupled to wind turbine installations. Wind turbine speed fluctuations can be tracked and utilized to control reactive and active power, according to the authors in the reference [93]. As referred to [94], state-of-the-art non-linear predictive model control (NMPC) techniques have been used to derive the highest quantity of power from the accessible wind speed. Also, state the efficiency and performance of the NMPC algorithm are higher than other widely used methods such as linear-quadratic Gaussian approaches, parameter ramps, proportional integration differentiators, etc. In addition, the authors in [95] state and justify that the performance and efficiency of the NMPC algorithm are higher than other widely used methods, such as linear-quadratic Gaussian approaches, parameter ramps, proportional integration differentiators, etc. S. Sabzevari et al. [96] demonstrated that model-based predictive control can operate smoothly in turbulent and wind-influenced modes and that a significant improvement has been achieved with the model-based predictive control concept. According to [90], model-free predictive control strategies for wind energy conversion systems based on a dual-fed machine tied to the network are superior to other conventional algorithms in terms of accuracy, simplicity, and efficiency. Figure 10 shows the fundamental concept of predictive model control (MPC) [97].

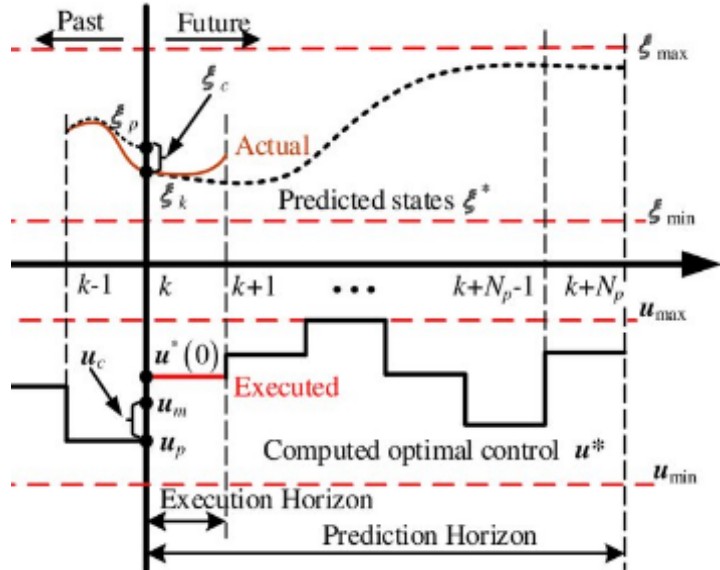

**Figure 10.** The fundamental concept of MPC [97].

### *4.6. Adaptive Controllers*

Building an unknown non-linear dynamic model of a WT system is extremely difficult, especially when adaptive control is considered. The adaptive control approach has been used for wind energy conversion systems because of the severe intrinsic nonlinear characteristics. According to [98], many direct adaptive control systems have been developed. In referring to [92], the author's attention was drawn to two different control strategies: one way uses preliminary estimations of wind turbine system non-linearities as a basis for supervisory control; the other uses unique dynamics in the wind power conversion system to eliminate tracking error and achieve a radial system. The authors concentrated on the direct adaptive control technique in a way that differs from the authors' focus on wind speed tracking, as they sought to achieve asymptotically stable turbine speeds by optimizing their wind speed measurement tracking control instead of focusing on wind speed tracking [92,93]. Lyapunov approach of self-tuning proportional–integral–derivative (PID) controllers for wind turbine systems has been researched and studied in [94]. The authors proposed this control method in which the limitations of the PID controller of the wind turbine dynamic system were first evaluated and upgraded by learning the impulse response filter. As referred to [95,96], the adaptive control method based on the idea of Lyapunov is more appropriate. It responds better to wind turbine changes than the self-regulating control strategy.

### *4.7. Robust Controllers*

According to [12,99–102], many researchers have developed a stable controller because of the inconsistency of wind speed in power systems. The aforementioned publications suggested feedback loop techniques may be used to determine maximum power and reduced load variation. In this regard, a robust control solution for the optimal power output of the VSWT is proposed in [103]. The results of this study showed that the proposed controller increased the WT energy output, estimated to be in the range of 15% to 20%, compared to a WT with the same rotor and a fixed transmission. In Ref. [104], robust control of a twin WTs structure based on a sliding mode controller is designed to track the maximum power by controlling the rotor speed and the yaw rotation but without the yaw actuator. The performances of the proposed control strategy are compared to the standard proportional-integral controller and show better results.

Two of the best advance and robust controllers are $H_2$ and $H\infty$. Both controllers are advanced robust-control strategies used for linear systems considering a bounded range of

external disturbances plus modeling uncertainties. These two controllers are used in wind energy systems to ensure a high-quality power supply. For instance, R. Rocha et al. [105] improved that it is necessary to compare the efficiency of H-infinity and H-2 controlled in terms of wind energy conversion systems to justify the difference. When compared to H-2 control principles, the H-infinity controller is shown to provide a more robust mechanism and have the shortest reaction time possible. Finally, the findings lead to the conclusion that neither the H-infinity controller nor the H-2 controller is appropriate for use with WT energy systems that vary in speed. The wind turbine systems' internal organization may benefit and be made easy by the power control approaches. In light of these ambiguous circumstances, the authors in [106] state that adapting the induction machine restrictions while still maintaining an efficient response to an unknown wind speed has been a focus of research for wind turbine systems using a linear time-invariant function. Robust control of variable-speed wind turbines based on an aerodynamic torque observer is proposed in [107]. This study showed that the proposed control policy is effective in terms of optimal power extraction and is robust with respect to uncertainties affecting the system.

Many researchers [108,109], proposed gain programmed controllers using estimates of wind turbine torque at their maximum operating points, which are provided by anemometers and Kalman filters (KF). From the available literature, it can be safely concluded that this controller exhibits high efficiency, has the ability to compensate for uncertainties and provide system stability. However, this controller's drawback is its control scheme's complexity. An increase in the mechanical stress of the WT as a result of sudden changes in control variables is another major setback for the pitch controller. It is necessary to state that robust controllers depend on the prior insight of the WT system and its mechanical model. The robust controller integrates feedforward, feedback system and sliding mode control to improve the robustness of pitch angle control in WT systems [12].

### 4.8. Optimal Controllers

Many scientists have put forward their own theories on how to get the most out of wind speed as well as the optimal approach for managing WTs under various wind conditions has yet to be found [110,111]. The optimum control techniques for MPPT wind power conversion systems employing fixed-pitch permanent magnet synchronous machines were subsequently advocated in [112]. The authors combined the DC voltage characteristics vs. maximum projected DC power and the stator frequency of permanent magnet machines using this optimum control technique to enable the wind power conversion system to function at its maximum power output. No further methods, such as wind speed monitoring, were required to determine the optimal power level of wind turbines. Refereeing to [113], which claimed that the fast flourier transform might be utilized to find the point of greatest power between the rotational speed and the dimensionless power factor derived from the calculated power. The direct drive control technique, which is innovative and straightforward for permanent magnet synchronous motor (PMSM) using a variable speed WT, is presented in [114]. Under constant and variable load situations, simple and less expensive vector control concepts were used to regulate frequency and voltage under unpredictable wind conditions to monitor the peak power point of a small WT system under distant power supply locations.

### 4.9. Neural Network Controllers

The maximum power of an artificial neural network (ANN) can be tracked in both stationary and dynamic situations, as well as a wind speed tracking system that is faster than an anemometer [115]. This method can be used to construct hardware, so a digital controller is not necessary. The best wind speed rotation for MPPT under unpredictable wind speed conditions was predicted using an ANN [116]. It was suggested that an ANN be utilized to define the reference tracking speed of the rotor using four input signals: rotor speed, output power, wind speed, and ideal power. The study's findings show that an effective ANN control method for wind energy conversion systems with PMSM has

been developed. Wind turbine system performance may be improved using a model developed by researchers using Markov and ANN [106,117]. With these strategies, it has been demonstrated that generator speed fluctuations are reduced and increasing the safety of the WT system and the ability to generate more electricity from varying wind speeds.

### 4.10. Fuzzy Logic Controllers

There has been a lot of interest in understanding-based control strategies for WECS. As it is well-known, fuzzy logic control (FLC) in wind turbine systems has been studied only up to this point. FLC techniques can be used to track the speed of WT using the cyclo-converter concept [108]. The authors in [109] have developed an FLC to extract the best wind speed and control unknown wind speeds. The developed techniques were tested on real metrological data. The results of the suggested fuzzy design show that by utilizing an 800 kW wind turbine system, Fuzzy controllers can enhance wind turbine performance at all wind turbine speeds, including the lowest, rated, and maximum speeds. An FLC is used in [118] and showed good performance for wind speeds below and over the rated wind speed. Using bilinear matrix inequalities, the authors created a fuzzy controller based on H-infinity methods [118]. The two-step approach of LMIs [118,119] is reduced in this controller class, which may be evaluated efficiently utilizing the convex optimization principle.

### 4.11. Unified DFIG Control

A hierarchical control system governs the operation of wind turbines; global control goals are achieved by modifying inputs to subordinate control loops. The operational characteristics of the dynamic and stable states guide the allocation of control duties. A representation of the global control system is shown in Figure 11. The rotor speed, reactive power, dc-link voltage, energy produced, and the stator flux size are among the controls that the unified DFIG WT architecture aims to achieve [120]. Internal control loops on the SGSC and MSC govern stator flux and the rotor current, respectively.

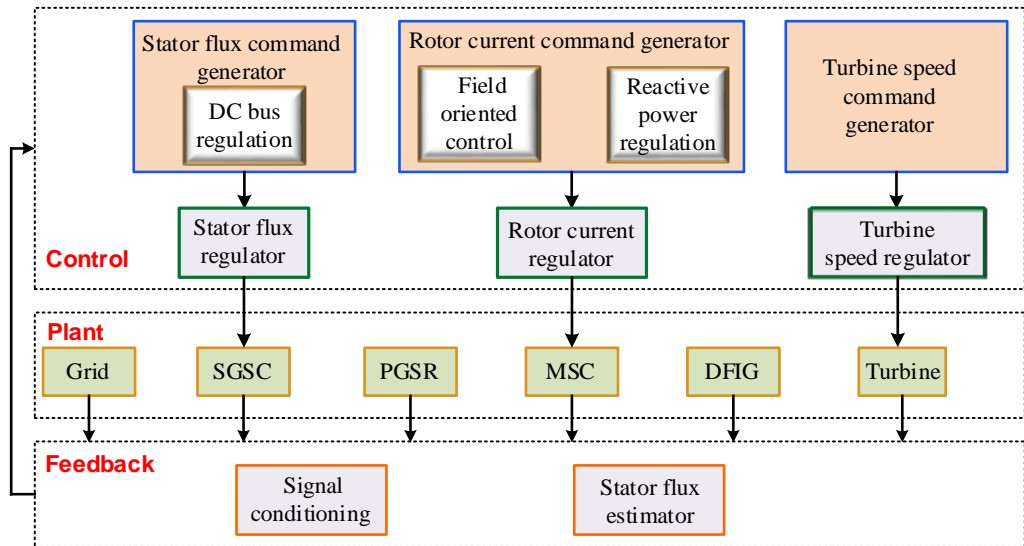

**Figure 11.** Fundamental concept Unified DFIG wind turbine layout with the global control structure.

In order to manage the MSC's current flow, torque, and wind power plant collector reactive power control loops must be used. Stator flux regulation commands are derived via the outer loop dc link voltage and dc link flux magnitude regulators. The SGSC and PGSR will swap power processing duties throughout the transition from sub- to super-synchronous operation; thus, these two outside loops supplying the flux command are flexible enough to handle that [88]. Blade pitch actuators control rotor speed to reduce the throttle mechanical torque output and coefficient of performance [72].

For the turbine speed command generator and regulator, the pitching of the WT blades is utilized to regulate the turbine's rotor speed by reducing the mechanical torque produced (Figure 12). It is set to the greatest possible speed for each unit of measurement [110]. The rotor speed instruction is limited to a safe level by a saturation block. Proportional-integral regulator based on the rotor speed error is utilized to regulate the blade pitch angle. It is possible to significantly enhance the blade pitch responsiveness for long-duration sags by feeding forward voltage sag information [111].

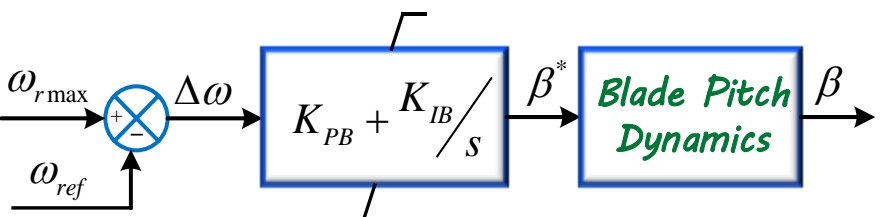

**Figure 12.** Turbine speed command generator and speed regulator.

*4.12. Machine and Grid Side Controllers*

The MSC controls variable speed operation to increase energy harvesting. The MSC is responsible for regulating the rotor speed to achieve the highest level of system stability and power. On the other hand, the GSC is responsible for controlling the DC link voltage of the converter in addition to the reactive and active power that is delivered into the grid, as can be seen in Figure 13a [40,112]. The rotor speed control and the response of DFIG grid-side T-type converters are displayed in Figure 13b,c [113].

Field-oriented control (FOC) and direct torque control (DTC) are two categories of MSC. Dynamic performance and characteristics of FOC and DTC are quite comparable [112]. For controlling generator speed, FOC uses a dual loop controller mechanism (outer loop and inner loop). The outer loop control needs rotor position and speed to produce a reference current for the three phases. Natural or synchronous reference frames are commonly used as the basis for the control of intra-loops [114]. For optimum electromagnetic torque with the smallest amount of stator current, the d-axis current of the stator is zeroed out [121], and the stator current q-axis is used to regulate the produced electromagnetic torque [122]. Due to the FOC's direct current management, more of the machine's ideal efficiency is used for torque output. Faster reaction and less complexity can be achieved with direct torque and power control through the DTC system [123]. Double-loop DTC is no longer necessary. A convertible switching pulse is generated directly from the hysteresis compensator output and flux angle [124]. To evaluate the performance of DTC, it is necessary to take into account the torque and current ripples. The dynamic characteristics of both controllers are identical [125]. The DTC controllers provide benefits, including getting rid of the rotor speed sensor, not having a current regulation loop, and having quicker reaction times. The main disadvantage of a DTC controller is the need for variable switching frequency. The kind of generator converter connected to the system has no bearing on the GSC. The grid integration of WECS is primarily the focus of GSC. GSC is classified into the direct power control (DPC) and voltage-oriented control (VOC) categories [126]. Because both VOC and FOC use a dual-loop control mechanism, they may be compared. When using the VOC technique, you'll have the option of using either PI or hysteresis-based control in either a synchronous frame of reference or a natural frame of reference for your internal current and DC link voltage control loops. When the q-axis is positioned so that it reads zero for the given reference, it is possible to realize the unity power factor [126]. VOC has a high steady-state capability, quick response time, superior power quality, and low power ripple. Their disadvantages are the reactive and active components of the VOC, as well as the need for a reference system [127]. A comprehensive evaluation of the MSC/GSC management approach is shown in Table 1 [128–131]. The controller based on VOC and FOC provides a suitable presentation with high efficacy for network incorporation, according to the aforementioned studies.

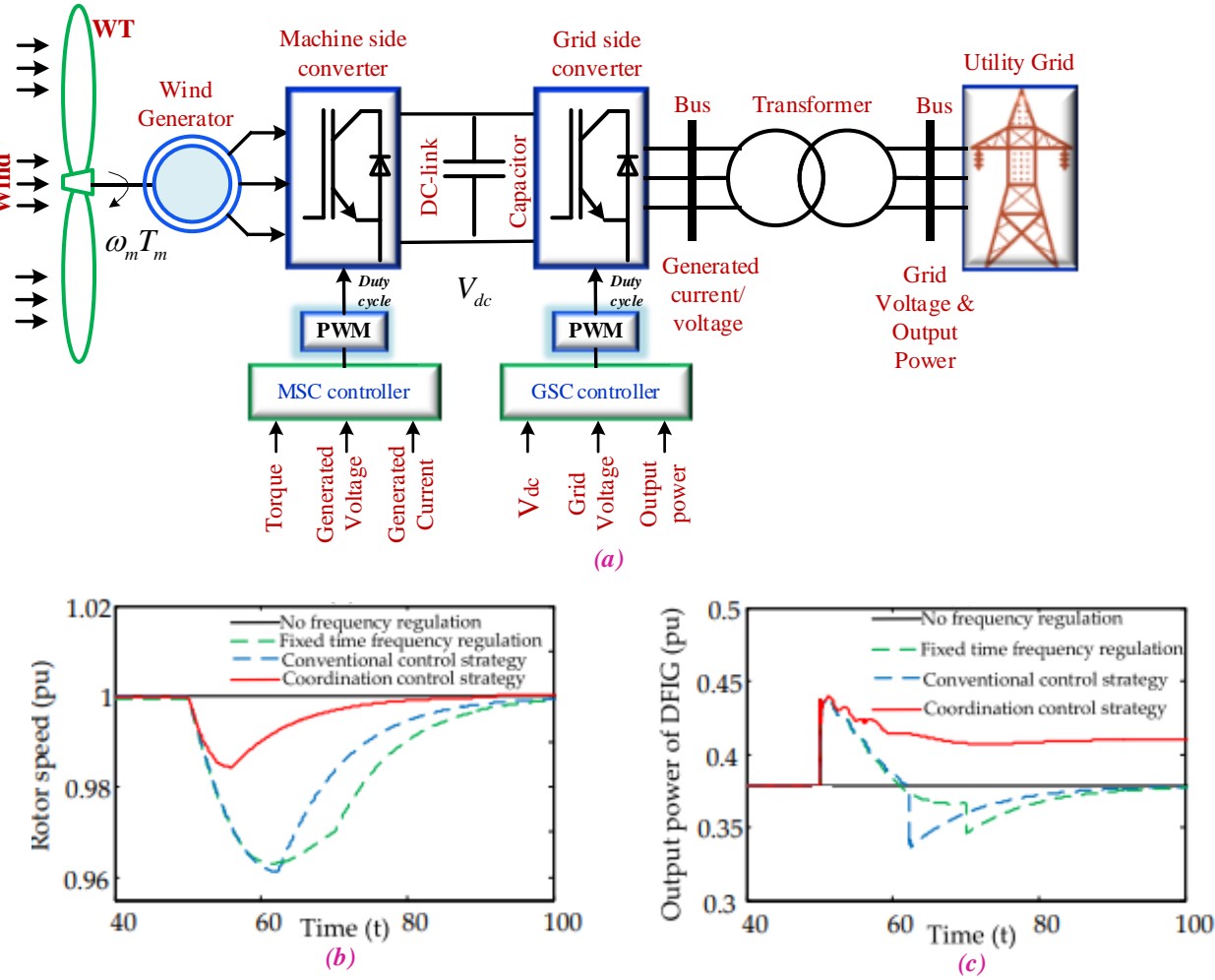

**Figure 13.** (**a**) MSC- and GSC-based controller for WECS; (**b**) Rotor side control [113]; and (**c**) Grid side of DFIG active power [113].

**Table 1.** A comparison of the MSC-based and GSC-based techniques.

| Parameter | GSC | | MSC | |
|---|---|---|---|---|
| | **DPC** | **VOC** | **DTC** | **FOC** |
| Implementation | Simple | Complex | Simple | Complex |
| Dynamic Response time | Low | High | Low | High |
| Power quality | Poor | Better | Poor | Better |
| Coordinate transformation | Not Required | Required | Not Required | Required |
| Power and current ripple | More | Less | More | Less |
| Internal current regulation loop | Not Required | Required | Not Required | Required |
| Power quality | Poor | Better | Poor | Better |
| Parameter Sensitivity | Insensitive | Sensitive | Insensitive | Sensitive |
| Rotor position sensor requirement | - | - | Not Required | Required |
| Torque ripple | - | - | More | Less |
| DC-link Voltage ripple | High | Low | - | - |

### 4.13. MPPT Controllers

WECS needs to have an MPPT algorithm so that the highest feasible power can be harvested from the wind, which changes dynamically with wind speed. The greatest amount of electricity may be extracted from a generator at a certain speed, depending on the wind speed [24]. Above a certain generator speed, the amount of electricity generated decreases significantly. Consequently, the variable speed wind turbine uses an MPPT controller to monitor the precise speed and harvest the highest quantity of power [40]. As shown in Figure 14a, the MPPT controller is mainly used in the second working area. The wind turbines in the second zone tend to harvest the highest amount of electricity [132]. In the third section, the power production is stabilized by lowering the mechanical speed to avoid causing damage to the WT and generator [133]. Figure 14a depicts the WECS MPPT topology in its most basic form of MPPT [24,39,40]. For example, the authors in [134], proposed an MPPT controller using the direct and indirect methods for WECS. Figure 14b displays the simulation results of the proposed MPPT method by modeling a range of wind speeds [134].

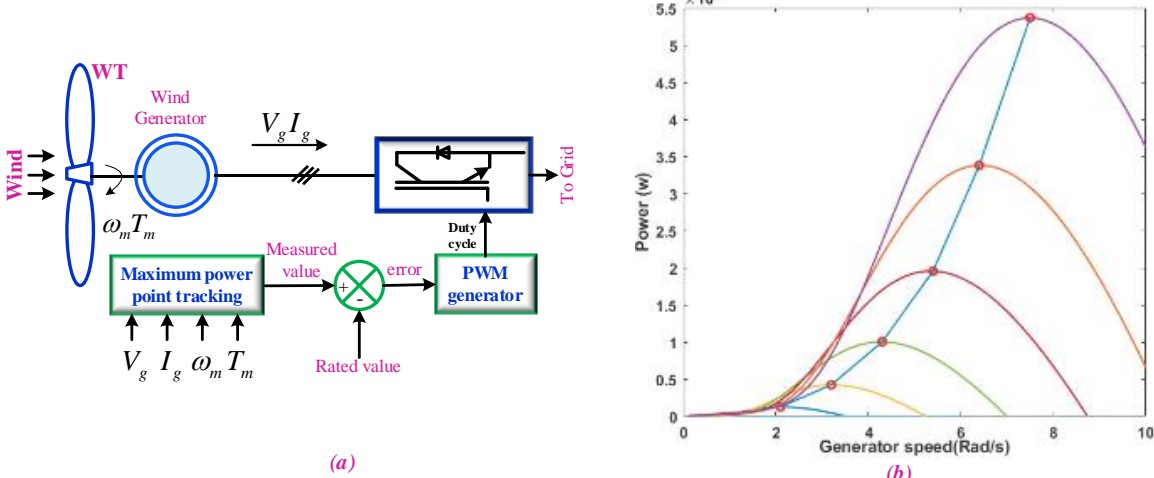

**Figure 14.** (**a**) Control of the conversion of wind energy based on MPPT and (**b**) Keeping track of the maximum wind power during a series of wind speeds [134].

Advanced MPPT algorithms that are widely used are called perturb and observe (P&O) or hill climb search (HCS) [135–137], power signal feedback (PSF) [138,139], tip speed ratio (TSR) [140,141], an optimum torque controller and other soft computing methods such as artificial neural network (ANN) and fuzzy logic controller (FLC) [142,143] and a hybrid of the above controller [144–146]. Various sensorless solutions have recently gained popularity due to the lack of anemometers and other costly sensors and effective accuracy with a rapid switching rate. Table 2 compares the various types of MPPT controller techniques related to WECS. There must be an understanding of the system prior to using the power signal feedback method. It is entered into the lookup table instead of measuring maximum power and shaft speed [147]. The most sophisticated PSF method records DC voltage and DC [133,148]. The lookup table parameter and available wind speed have a connection, and this relationship is used to calculate the optimum power. The system's most significant downside is its complicated implementation [149]. An MPPT approach commonly employed is HCS or P&O. This method alters input voltage or duty cycle, for example, by looking at the previous cycle's output power and generating the necessary step size for the next one [135].

**Table 2.** Different MPPT controller in wind energy: A comparison.

| Techniques/Parameter | TSR | SCT | HCS | OTC | PSF | Hybrid |
|---|---|---|---|---|---|---|
| Efficiency | Very High | High | Low | Moderate-High | Moderate | Very High |
| Complexity | Simple | - | Simple | Simple | Simple | - |
| Convergence speed | Fast | Medium | Low | Fast | Fast | Fast |
| Wind speed measurement | Required | Required | Not Required | Not Required | Required | Not Required |
| Prior knowledge | Required | Not Required | Not Required | Required | Required | Depends |
| Tolerance to rapid variation | Moderate-high | High | Low | Moderate-high | Moderate | Very High |
| Sensitivity | No | No | No | Yes | Yes | Depends |
| Memory requirement | Not Required | Not Required | Not Required | Depends | Required | Depends |

## 5. Converter Applications

More and more industries and applications are relying on converters because it is becoming more cost-effective, dependable, flexible, and easy to integrate into various systems. Converter applications on energy saving, electric vehicles (EV), renewable energy systems, and future sustainable technologies are also discussed.

### 5.1. Energy Savings

Reducing the amount of electricity generated by fossil fuel-based power plants and hence lowering pollution levels is one way to save energy [150]. As previously stated, converter devices are naturally more efficient in general energy processing applications than other types of devices. It is time to talk about a few more energy-saving ideas. A considerable portion of grid energy is used to power electric motor drives in the United States and other Western nations (60% or more), with pumps, fans, and compressors accounting for the majority of these (usually 75%). A 20% reduction in energy use under a light load may be achieved by switching to a motor speed controlled by a variable frequency instead of a constant voltage motor speed [18].

### 5.2. Electric Vehicles

Research and uses of electric and hybrid vehicles are primarily motivated by the oil shortage and environmental pollution management. Energy is also conserved when EV/HEVs replace internal combustion engine cars. However, the power used to charge a battery must come from a clean, renewable source such as a wind power system. Alternatively, in case of fossil fuels are used to create energy, urban pollution is transmitted to the power plant. Similarly, an electric vehicle powered by renewable energy or fuel cell may be utilized to create electricity that generates hydrogen gas as fuel. Figure 15 depicts an EV driving system [151], in which the battery serves as the energy storage device. An IGBT-based PWM inverter converts the direct current to variable voltage power and variable frequency that powers an internal permanent magnet (IPM) synchronous motor. The IPM machine features improved performance, a smaller footprint, and a large field-weakening area for optimal control of speed. The primary reason why EVs consume less energy is because of regeneration, which recovers braking energy.

### 5.3. Converter-Interfaced Renewable Energy Systems

A significant amount of the world's energy demand may be fulfilled by encouraging environmentally friendly renewable sources [152–154]. Currently, the entire planet is going in that direction. Unlike fossil and nuclear energy sources, RES is not exhausted as they are used. The primary renewable energy sources include hydropower, solar, wind, biofuels, wave, geothermal, and tidal, which are plentiful and ecologically friendly. It is noteworthy to note that 22.9% of electrical energy in the US is presently generated from renewable sources, [155] which is greater than the proportion of nuclear power. There is currently hope for a 100% renewable power system in the US by 2050 based on current trends [156]. Denmark has set a high bar for itself, aiming to generate all of its power and heat from

renewable sources by 2035 and eliminate the use of all fossil fuels by the year 2050 [157]. The authors in [158], expected that renewable energy sources may meet the world's total energy demand provided storage and transmission capacity are enough.

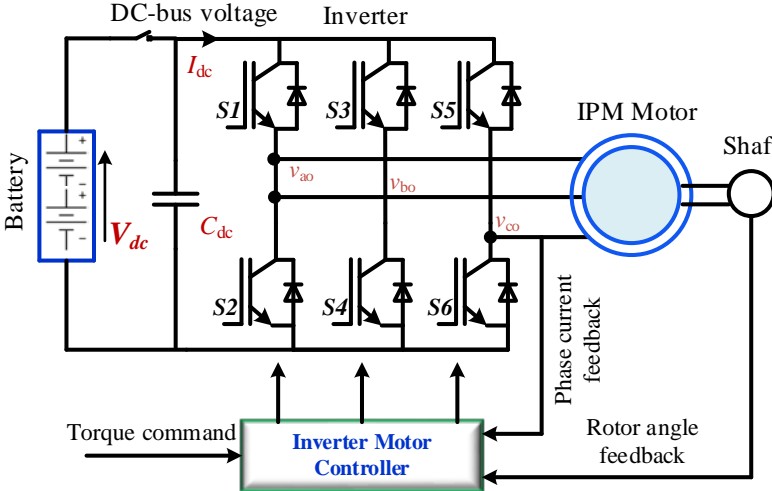

**Figure 15.** Schematic representation of the IPM machine (GEETXII) being used to power an electric vehicle.

### 5.3.1. Wind Energy Systems

Figure 16a depicts a typical wind generating system in which a speedup gear connects the shaft of an ac machine to a wind turbine with variable speed along with conversion devices [159]. Before feeding it to a grid using a step-up transformer, PWM converters convert the voltage and frequency variable values to a constant value. Alternatively, it can produce a self-sufficient load for itself. The alternating currents on the ac sides have a sinusoidal waveform and a power factor that can be adjusted. As depicted, the control system detects the arbitrary wind speed and controls the generator speed to optimal energy production. MPPT allows for the identification of the maximum power point. By coordinating the line current and phase voltage, the line-side converter is able to keep the dc link voltage under control and hence regulate it. The design depicted in Figure 16a applies equally to four-quadrant industrial drive systems.

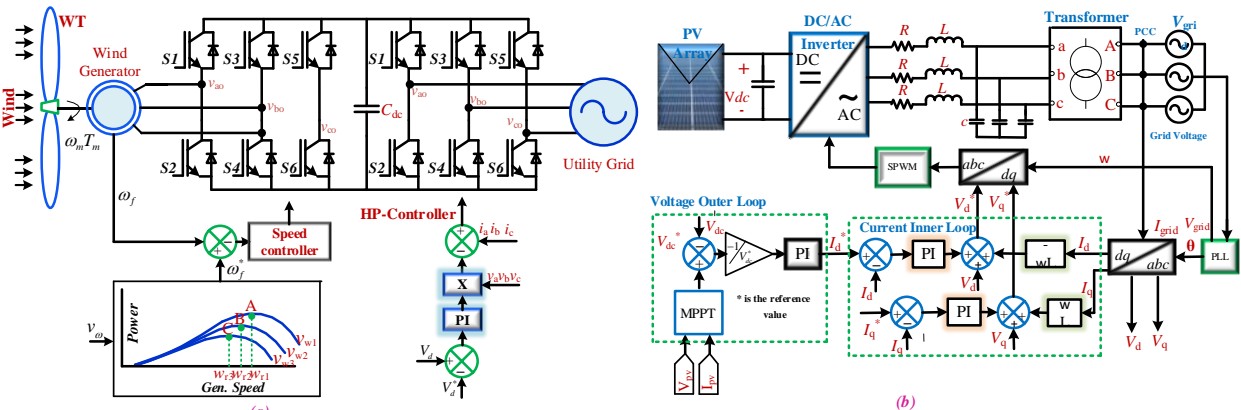

**Figure 16.** (**a**) Wind power generation system utilizing an induction generator and power electronics conversion, (**b**) Photovoltaic power conversion system.

### 5.3.2. Photovoltaic Systems

PV systems and solar thermo-electric systems are the two main categories of solar electrical energy [160]. In the first scenario, mirrors focus solar heat on turning water into

steam. This steam then powers a turbo-generator system, which generates energy. These solar cells (Si, CdTe, CdGe) turn sunlight directly into energy when used in a photovoltaic system (PV). In either case, the produced dc is subsequently converted to ac and supplied into the grid or utilized to power the system. With a PV array (series-parallel cell connection) coupled to a DC-DC converter for voltage boost and an AC inverter (PWM), this is a typical PV system setup shown in Figure 16b. The ac grid is fed by transformers that connect many units. Using the MPPT search method, the dc–dc converter regulates the array's maximum available power output. Applications that necessitate a large amount of electricity can use converters at many levels. Inverters allow users to control both active and reactive power [161].

*5.4. Future Wind Energy Converter Technologies*

Future research on wind systems will mainly be based on how well the system connected to the grid performs in fault recovery (FRT) mode. New ground has also been broken in the area of WECS concerning the gathering network for offshore wind power installations using PE devices [15]. The input parameters for each transducer can be minimized with a multi-purpose tilt controller and an MPPT controller. This is also an excellent way to plan for future controllers [116]. Most wind turbines being constructed today rely on power conversion that occurs at low voltage sides. In order to accommodate the rapidly expanding capacity of wind farms and wind turbines, it is projected that new converters technologies will be developed that will be capable of providing more efficient and reliable power conversion at greater voltage levels (1–10 kV) in the near future [162]. There has been a significant amount of improvement made to converters and semiconductor devices. Because of their ability to convert greater voltages and powers, multilevel converters may soon replace full-scale power converters as the most popular option for WTC that are based on power conversion [59]. Figure 17a shows how the three-level active/non-active neutral-point diode clamped (3L-NPC/ANPC) converter that applied in different wind power applications. One of the multilevel topologies utilized most frequently on the market today is the 3L-NPC/ANPC. This type of converter reaches one additional voltage level and reduces dv/dt stress compared to the 2L-VSC converter; consequently, medium-voltage power may be generated [163,164]. Figure 17b shows the converter system that utilizes a solid-state dc transformer designed for use in various applications of wind power [163,164].

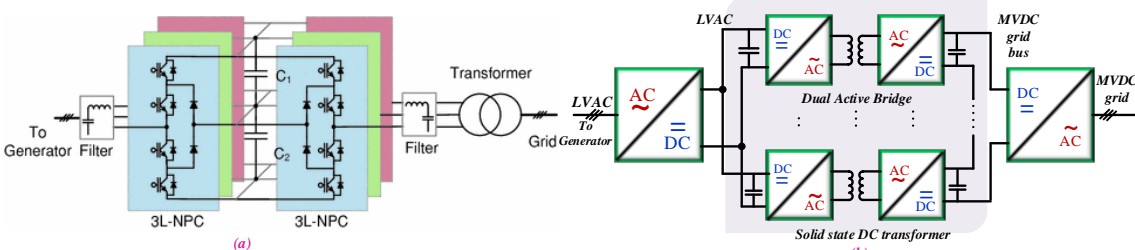

**Figure 17.** (**a**) Converter (3L-NPC BTB) used for wind turbines [164]; (**b**) A converter system that utilizes a solid-state dc transformer is designed for use in wind power applications.

Based on the European UNIFLEX-PM project and the American FREEDM project [81], the WTC might also benefit from an improved converter design that has a similar idea to the traction converters of the future. There are many dual-active-bridge (DAB) building blocks with galvanic isolation, as shown in Figure 17b, which is the basis for this solid-state dc transformer. Due to medium-frequency excitation, the size and weight of the transformer within the DAB may be reduced. Furthermore, in the future, a medium-voltage dc/ac converter may be used to connect the solid-state dc transformer directly to the medium-voltage dc distribution network or the ac distribution network (10–20 kV).

## 6. Conclusions and Future Directions

The use of converters and the accompanying control strategies is becoming increasingly important in the process of efficiently harvesting energy from wind. They are significant contributors to the process of energy conversion. The fact is that wind energy still suffers from low conversion efficiency. Thus, the development of efficient converter devices along with robust controllers will ensure a supply of power that is both high quality and reliable. As a result of the growing penetration of wind turbines into the electrical system, It is required to produce power effectively to comply with the grid code. To achieve this target, many converter devices have been developed for the wind energy conversion system. In this regard, a comprehensive review of the role of converters for wind power systems in terms of energy conversions, controls, and applications was highlighted in detail. In this study, the authors provided a thorough assessment of converters for the integration and control of wind turbines. Additionally, they investigated the functioning and application of control for the wind energy power system. In the future, the application of advanced converter devices may lead to a more reliable generation of power as well as a reduction in the overall cost of the system. Based on this review, the following is a description of some components that are credited with contributing to the positive prospects for the future of systems that convert wind energy.

- More accurate and precise physics-based motivated, dynamic forms of these constraints are now available, collected, and distributed across current and future wind energy conversion systems.
- The most significant advancement in the research study's future must be the inclusion of efficient control plan methods as a need for the assessment of sustainable technologies for converting wind energy.
- The development of different control systems for forecasting energy consumption is more crucial.
- For the integration of wind turbine systems, more sophisticated controller approaches of the current controller used in wind energy systems are needed, along with an industry-standard integrated, flexible control system that is durable, adaptive, and optimum.
- More sophisticated software applications to model, design, analyze, test, and validate the capability and adaptability of an AC-connected wind power conversion system and the Internet.
- It has been concluded that future studies' key objectives are focused on improving converters' application for wind systems in terms of security, cost-effectiveness, usability, compliance, monitoring, and sustainability.

**Author Contributions:** A.Q.A.-S. and M.A.H.; resources, M.S.M.; writing and original draft preparation, T.M.I.M., review and comment, M.M. (M. Mannan) and H.M.K.A.-M.; visualization, M.A.H. and T.M.I.M.; supervision and coordination, M.A.H.; project administration., P.J.K. and M.M. (M. Mansor); funding acquisition. All authors have read and agreed to the published version of the manuscript.

**Funding:** The Ministry of Higher Education, Malaysia supports this work under the long-term research grant scheme (LRGS) program project grant no. 20190101LRGS and HICOE wind project code 2022004HICOE under the Institute of Sustainable Energy, the Universiti Tenaga Nasional, Malaysia.

**Institutional Review Board Statement:** Note applicable.

**Informed Consent Statement:** Not applicable.

**Data Availability Statement:** No data were created or used.

**Conflicts of Interest:** The authors declare no competing interest.

## Abbreviations

| | | | |
|---|---|---|---|
| 2LG | double-line-to-ground | PMSM | permanent magnet synchronous motor |
| ANN | artificial neural network | P&O | perturb and observe |
| b2b | back-to-back | PV | photovoltaic system |
| DTC | direct torque control | PE | power electronics |
| DPC | direct power control | PSF | power signal feedback |
| DDSG | directly driven synchronous generator | MPC | predictive model control |
| DFIG | doubly fed induction generators | PID | proportional–integral–derivative |
| DAB | dual-active-bridge | PWM | pulse width modulation |
| EV | electric vehicle | RES | renewable energy source |
| FOC | Field-oriented control | RSC | rotor side control |
| FSWT | fixed-speed WT | SCIG | squirrel cage induction generator |
| FLC | fuzzy logic control | SMC | sliding mode controller |
| GSC | grid-side converter | STATCOM | static synchronous compensator |
| HCS | hill climb search | TSR | tip speed ratio |
| HAWT | horizontal-axis wind turbine | VSWT | variable-speed WT |
| IPM | internal permanent magnet | VSCI | variable structure integral |
| IVSC | integrated variable structure controllers | VAWT | vertical-axis wind turbine |
| MPPT | maximum power point | VOC | voltage-oriented control |
| NMPC | non-linear predictive model control | WECS | wind energy conversion system |
| PMG | permanent magnet generators | WT | wind turbine |
| PMSG | permanent magnet synchronous generator | WTS | wind turbine systems |

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
