# Peer review of "Wind Energy Conversions, Controls, and Applications: A Review for Sustainable Technologies and Directions"

_sustainability, doi:10.3390/su15053986_

Round 1
Reviewer 1 Report
This manuscript reviews the impact of converter operation, control, and application on wind energy conversion. The impact of current converters on the integration and control of wind turbines is described through a detailed overview of recent advances in converter control systems. This manuscript could be an addition to the review of wind energy converters. The manuscript needs a careful text editing and an improvement of the English language to better represent the results.
I suggest that the paper can be accepted after revisions. Some questions are listed below:
1. Page 1: Many abbreviations are not defined when they first appear, such as PWM (line 80), PMSG (line 140), PMG (line 158), RSC (line 370), STATCOM (line407), etc. It is suggested to summarize the abbreviations for the convenience of reading.
2. Line 107: The text in Fig. 2 is blurry. Please provide a clearer version.
3. Line 127: There is an error in the sentence. Please check it.
4. Line 144: Is there a more detailed quantitative description of the advantages of PMSG compared to DFIG?
5. Line 160: Please give detailed examples of material innovations for generator rotors to enhance credibility.
6. Line 267: The content of the entire section 4.1 is very similar to the content of section II.A in the literature (Z. Chen, J. M. Guerrero and F. Blaabjerg, "A Review of the State of the Art of Power Electronics for Wind Turbines," in IEEE Transactions on Power Electronics, vol. 24, no. 8, pp. 1859-1875, Aug. 2009, doi: 10.1109/TPEL.2009.2017082.). Please revise the contents of the section 4.1 or indicate the citation.
7. Line 295: The contents of Fig. 6(b) cannot be found in [68]. Please confirm that the citation is correct. The quantitative information of the peak currents in Fig. 6(b) is difficult to follow. Please provide a clearer figure.
8. Line 312: There is a grammatical error in the sentence.
9. Line 334: “100% 2LG fault” needs to be described in detail. In addition, there is an error in the citation of the literature [76].
10. Line 457: The ANN-SMC lacks a specific description.
11. Line 505: As a controller capable of ensuring a high-quality power supply as suggested in the conclusion (line 784), the review of robust controllers is not sufficiently deep and specific.
12. Line 510: The appearances of the concepts of “H-2” and “H-infinity” are abrupt. Please describe them in more detail.
13. Line 599: “Fig.13(a) and Fig.13(b)” should be changed to “Fig.13(b) and Fig.13(c)”. In addition, there is a citation error here.
14. Line 805: The recommendation is fuzzy and not specific. What does “sophisticated” refer to specifically?
15. Line 828: The format of some references needs to be standardized.
Author Response
Dear Reviewer,
Please find the attached reviewer response sheet where we have provided point by point responses on the reviewer comments and suggestions.
Thank you,
M A Hannan

Reviewer 2 Report
The author discussed the wind energy conversions, controls, and applications. Does the batteries have been employed to stroage the energy? What kinds of the batteries is the best choice?
Author Response

(The authors gave the same response as above.)

Reviewer 3 Report
After reading and understanding the paper content titled as “Wind energy conversions, controls, and applications: A review for sustainable technologies and directions” the reviewer writes the following comments:
(1) In general:
The paper is a review paper, which summarizes information came out of relevant papers collected in the field mentioned in the title. As the review papers usually do, authors collect all relevant and determinative results, progresses, news belonging to the worked topic. This paper is a good review paper. This paper focuses on wind energy generation and its aspects. The paper introduces the process for electrical energy generation originating from blowing wind. In other words, it introduces all the parts play important role at the energy transfer, where the kinetic energy of the moving air or blowing wind transfers to electrical energy form going through many steps. The paper - as part of the energy transformation - also talks about wind energy conversions, controls and application possibilities. This paper processes many determinative results worldwide at the field wind energy generation bordered by the not too far past and present as opened brackets of the time interval. The Authors started collecting 533 relevant papers, and after their filtering processes 287 papers put in the focus. Finally, 172 papers were selected and used to write this review paper based on them. I think, the 172 selected papers are good quality scientific publications, which are/were milestones at the actual research field.
(2) Length and elaboration of the paper:
According to the reviewer, the length of the text is appropriate to the review type papers. In the draft version of the paper, it consists 29 pages, 24 pages scientific text and additionally 5 pages for the References. Structure of the paper follows the general and traditional way for review papers. Introduction leads up the content well. As nexts steps, the main parts of the text are divided into chapters and subchapters. Structure of the chapters and subchapters are well built up. Every subchapters collects and informs important says from the actually referred papers.
(3) English language, grammar and spelling mistakes:
In my opinion the paper was written in good English style. It can feel, that the Authors speak and write good in English. Probably, in some cases are spelling mistakes in the paper, for example such as line 680 and 787: instead of the written “converters devices” the “converter devices” might be suggested. It is suggested to read and correct incidental spelling mistakes.
(4) Figures:
In the reviewer’s opinion quality and elaboration of all Figures are very good in the paper. I really like every Figure in the article. Figures are clear, understandable, really colorful and types and thicknesses of lines used are absolutely help the overview and survey of the actual Figure. Drawing technique of the Authors is great. Very insightful (picturesque) technique was used. It is enjoyable to look at and survey all Figures.
(5) Tables:
Table 1 and 2 are good. These tabular 1 and 2 collects and shows the most important and dominant scientific results belonging to the field discussed. In case of review type papers it is a practice inserting this type of Tables showing the relevant summary and collect every results born at the subfield in the not too far past and nowadays.
(6) Professionalism:
Professionalism of the actual paper is good. Reading the paper, reader may feel that Authors know the field of science what they are working on. Usage of the scientific wording is good. Authors know the vocabulary of the field very well. Abbreviations are also good and authors use the abbreviations smartly and well during the whole paper. Reader may feel that, it is a good review paper, paper radiates the professional safety.
(7) Readability of the paper:
As it was mentioned before, the paper is a well written text. Reviewer only has one negative remark here: Reading the paper text continuously it can feel, that the Authors were tired at the end of the paper writing. At the beginning of the paper elaboration and length of the subchapter texts are good and long enough (length of the subchapters are great), but towards the end of the article the elaboration gets a little worse, and the length and detail of the subchapters and also getting little worse (subchapters are getting shorter and shorter). Probably completing those texts parts with thoughts in a relaxed state could improve more the quality of the paper.
(8) Summary:
I think, it is a good review paper. It meets the requirements what the general review paper usually holds. Scientifically the paper collects the determinative information belongs to the field from the not too far past and present. Quality of the figures and figure captions are outstandingly good. Elaboration of the paper is good. One negative remark: It would improve the quality of the paper if the Authors could write more details and collect and insert more relevant information to the second half or to the end of the paper main text (complete with more information text). Towards the end of the paper text, contents are little short, they are written in brief only. A more thorough explanation of the topic at the short subchapters may be good.
After doing the suggested improvements paper is recommended for publication.
Author Response

(The authors gave the same response as above.)

Round 2
Reviewer 2 Report
The review can be published now.